# Medicinal Phytocompounds as Potential Inhibitors of p300-HIF1α Interaction: A Structure-Based Screening and Molecular Dynamics Simulation Study

**DOI:** 10.3390/ph18040602

**Published:** 2025-04-21

**Authors:** Muhammad Suleman, Abrar Mohammad Sayaf, Sohail Aftab, Mohammed Alissa, Abdullah Alghamdi, Suad A. Alghamdi, Mohammed A. Alshehri, Kar Kheng Yeoh, Sergio Crovella, Abdullah A. Shaito

**Affiliations:** 1Laboratory of Animal Research Center (LARC), Qatar University, Doha P.O. Box 2713, Qatar; m.suleman@qu.edu.qa; 2Center for Biotechnology and Microbiology, University of Swat, Swat 19200, Pakistan; sohailkhanpharmacist@yahoo.com; 3School of Chemical Sciences, Universiti Sains Malaysia, Gelugor 10050, Penang, Malaysia; amsayaf@gmail.com (A.M.S.); kkyeoh@usm.my (K.K.Y.); 4Department of Medical Laboratory, College of Applied Medical Sciences, Prince Sattam bin Abdulaziz University, Al-Kharj 11942, Saudi Arabia; m.alissa@psau.edu.sa (M.A.); ab.alghamdi@psau.edu.sa (A.A.); s.alghamdi@psau.edu.sa (S.A.A.); m.alshehri@psau.edu.sa (M.A.A.); 5Biomedical Research Center (BRC), Department of Biomedical Sciences at College of Health Sciences, and College of Medicine, QU Health, Qatar University, Doha P.O. Box 2713, Qatar

**Keywords:** p300, HIF-1, hypoxia, phytocompounds, drug screening, MD simulation, binding free energy

## Abstract

**Background:** Hypoxia plays a key role in cancer progression, mainly by stabilizing and activating hypoxia-inducible factor-1 (HIF-1). For HIF-1 to function under low oxygen conditions, it must interact with the transcriptional coactivator p300, a critical step for promoting cancer cell survival and adaptation in hypoxic environments. **Methods:** Consequently, we used drug design and molecular simulation techniques to screen phytochemical databases, including traditional Chinese and African medicine sources, for compounds that could disrupt the p300/HIF-1 interaction. **Results:** In this study, we identified potential compounds with high docking scores such as EA-176920 (−8.719), EA-46881231 (−8.642), SA-31161 (−9.580), SA-5280863 (−8.179), NE-5280362 (−10.287), NE-72276 (−9.017), NA-11210533 (−10.366), NA-11336960 (−7.818), TCM-5281792 (−12.648), and TCM-6441280 (−9.470 kcal/mol) as lead compounds. Furthermore, the compound with the highest docking score from each database (EA-176920, SA-31161, NE-5280362, NA-11210533, and TCM-5281792) was subjected to further analysis. The stable binding affinity of these compounds with p300 was confirmed by Post-simulation binding free energy (−22.0020 kcal/mol, −25.4499 kcal/mol, −32.4530 kcal/mol, −33.9918 kcal/mol, and −57.7755 kcal/mol, respectively) and KD analysis. Moreover, the selected compounds followed the Lipinski rules with favorable ADMET properties like efficient intestinal absorption, high water solubility, and no toxicity. **Conclusions:** Our findings highlight the potential of natural compounds to target key protein–protein interactions in cancer and lay the groundwork for future in vitro and in vivo studies to explore their therapeutic potential. Specifically, disrupting the p300/HIF-1 interaction could interfere with hypoxia-driven pathways that promote tumor growth, angiogenesis, and metastasis, offering a promising strategy to suppress cancer progression at the molecular level.

## 1. Introduction

Hypoxia, characterized by an abnormal oxygen level, is frequently observed in malignant tumors. It triggers the development of dysfunctional vascular networks within tumors and promotes the transition of cancer cells from an epithelial to a mesenchymal state, facilitating their mobility and metastasis [1]. Furthermore, hypoxia disrupts cancer cell metabolism and promotes resistance to therapy by inducing a state of dormancy [2]. Cycling hypoxia enhances the production of reactive oxygen species (ROS), which support the survival of tumor cells. In vivo and in vitro studies have shown that both short and long-term exposure to hypoxia increases the resistance of cancer cells to radiation therapy [3]. Whether during development or adaptation to low oxygen levels, hypoxia-inducible factor-1 (HIF-1) serves as a pivotal heterodimeric transcription factor, directing the expression of target genes [4]. The activation of HIF-1 can have either beneficial or detrimental roles in various pathological conditions [5,6]. For instance, in cardiac and cerebral ischemia it facilitates revascularization, yet in hypoxic environments, it fosters tumor growth by stimulating angiogenesis and metabolic adaptation to hypoxia. The stability and transcriptional efficacy of HIF1α rely on the recruitment of the transcriptional coactivator and integrator p300 [7].

In recent years, p300, also known as EP300 (E1A-associated protein p300) or KAT3B, has emerged as a crucial driver of oncogenesis. p300 and CBP (CREB binding protein, CREBBP, or KAT3A) exhibit high sequence similarity, structural organization, and overlapping yet distinct functions, which is why they are commonly referred to as CBP/p300. p300 and CPB are members of the KAT superfamily of proteins, and their primary function is to acetylate histones H3 and H4 [8]. p300 induces the acetylation of histone H3 lysine 27 (H3K27ac) at various genomic sites, including promoters, enhancers, and super-enhancers, thereby promoting gene transcription [9,10]. Moreover, studies have revealed that p300 is overexpressed in several types of cancer and drug-resistant cancer cells. This overexpression leads to the activation of oncogene transcription, fostering cancer cell proliferation, survival, tumorigenesis, metastasis, immune evasion, and drug resistance [11,12]. A study by Kung et al. suggests interplay between HIF-1 and p300, highlighting the specific binding of the CH1 domain of p300 to the C-terminal transactivation domain (CTAD) of HIF-1 [13]. By conducting a random mutagenesis screen, the study identified four critical residues in HIF-1—Leu-795, Cys-800, Leu-818, and Leu-822—and four in p300—Leu-344, Leu-345, Cys-388, and Cys-393—that are essential for the interaction between p300 and HIF-1 [7,14].

Research conducted in vivo has underscored the advantageous impact of disrupting the interaction between HIF1α and p300 in terms of suppressing both target genes and, consequently, tumor growth [15]. For instance, compounds such as chetomin have been shown to intervene in the HIF-1α/p300 interaction, resulting in a decreased expression of hypoxia-inducible genes. This subsequently leads to a reduction in tumor growth and angiogenesis. Additionally, the protein CITED2, known to bind with p300, is believed to exert a negative regulatory effect on HIF-1 transactivation [16]. The transactivation domain of CITED2 binds strongly to the CH1 domain of p300, disrupting its binding to HIF-1α. CITED2 competes effectively with HIF1α for binding to p300/CBP, showing a 33-fold higher affinity for the CH1 domain because of partially overlapping binding sites [17,18]. This interference by CITED2 inactivates HIF-1, suggesting its potential significance in cancer therapy [19]. Intensive research has focused on developing inhibitors targeting p300, with the aim of hindering its interaction with transcription factors and substrates by targeting its acetyltransferase domain. Despite exploring numerous synthetic drug candidates, challenges like liver toxicity and increased blood pressure have emerged. There remains an urgent need for p300 inhibitors with enhanced binding affinity, greater target selectivity, and reduced cytotoxicity to enhance their efficacy as antitumor therapeutics.

The diversity of natural products, distinguished by their unique molecular compositions, has historically served as a fertile ground for discovering pharmacological solutions across a spectrum of disease conditions [20,21,22]. Several natural and synthetic compounds have demonstrated the ability to modulate HIF-1α signaling, offering promising therapeutic potential against cancer. Chlorogenic acid has been shown to suppress angiogenesis by inhibiting the HIF-1α/AKT pathway [23], while sinometumine E, derived from Corydalis decumbens, promotes angiogenesis through the HIF-1/VEGF pathway in vivo and in vitro [24]. Graviola, a natural plant-derived compound, has exhibited significant antitumor effects by altering cell metabolism, thereby inhibiting the tumorigenicity and metastasis of pancreatic cancer cells [25]. Additionally, chetomin effectively attenuates HIF-1-mediated gene expression, suppressing tumor growth in human colon cancer and PC-3 models [26]. Novobiocin has also emerged as a novel agent that disrupts the interaction between HIF-1α and p300/CBP by directly binding to the HIF-1α C-terminal activation domain [27].

All of this considered, our study seeks to uncover novel inhibitors targeting the p300 enzyme by exploring a wide range of natural product databases. These databases, including the Traditional Chinese Medicines Database (TCM) [28], North African Natural Products Database (NANPDB) [29], East African Natural Products Database (EANPDB) [30], North-East African Natural Products Database (ANPDB) [31], and the South African Natural Compounds Database (SANCDB) [32], comprise natural products sourced from various origins, including plants, animals, fungi, bacteria, and marine organisms. This diverse pool offers a rich array of chemical structures and biological activities to explore new inhibitors of the p300 protein [33]. Therefore, we screened these natural products databases for p300 inhibitors that specifically target the interaction between p300 and HIF1α using the virtual drug screening approach. The molecular dynamic simulation and binding free energy calculations were used to shortlist compounds and validate their high binding affinity with p300. Overall, this research introduces new plant-derived compounds aimed at improving cancer treatment by disrupting the interaction between p300 and HIF1α.

## 2. Results and Discussion

The stability and transcriptional efficacy of HIF1α rely on the recruitment of p300 [7]. To screen phytocompounds deposited at the natural products databases (SANPDB, NANPDB, NEANPDB, EANPDB, and TCM) against p300, we retrieved the crystal structure of the p300 protein (ID: 1P4Q) from PDB [34]. The p300 structure was obtained from the complex of the p300- transactivation domain of CITED2 in the 1P4Q structure using PyMOL software (2.5.4) (Figure 1). Figure 1 illustrates the structural organization of p300, highlighting its key functional regions. The left panel shows the complex of p300-CITED2, with secondary structural elements depicted. The right panels further depict the E1A-associated protein p300 and the CBP/p300-interacting transactivator 2, emphasizing the domains involved in protein–protein interactions. In a previous study, the p300 residues Leu-344, Leu-345, Cys-388, and Cys-393 were identified as crucial for binding with HIF-1α. To inhibit the interaction between HIF-1α and p300, we targeted these key residues using a virtual drug screening approach with phytocompounds [14]. This screening identified 10 lead compounds (the two compounds with the highest docking scores from each database) with high docking scores to p300, ranging from −12.648 kcal/mol to −7.818 kcal/mol. Table 1 summarizes the docking analysis of the top 10 lead compounds identified from natural product databases as potential inhibitors of the p300-HIF-1 interaction. The table presents docking scores, key ligand functional groups, and their interactions with p300, including hydrogen bonding, hydrophobic interactions, and π-π stacking. The interaction distances (in Å) highlight binding strength and stability. Notably, the compounds NE-5280362, NA-11210533, and TCM-5281792 exhibited the strongest binding, forming multiple hydrogen bonds with key residues. These findings suggest promising candidates for further validation through molecular dynamics simulations and experimental studies.

### 2.1. Screening of EANPDB to Shortlist Potential Inhibitors of p300-HIF-1 Interaction

The virtual screening of the EANPDB identified the two natural compounds: 4′-O-methylepigallocatechin (EA-176920) and (+) Sesamin (EA-46881231), with docking scores of −8.719 kcal/mol and −8.642 kcal/mol, respectively (Table 1). 4′-O-methylepigallocatechin (EA-176920) is a flavonoid compound that has been isolated from the bark of *Elaeodendron schlechteranum*, a plant that is used in traditional medicine in Tanzania for treating various non-infectious diseases, such as septic wounds and boils [35]. This compound establishes interactions with the p300 active site via four hydrogen bonds and one pi–pi interaction. The hydrogen bonds occur between the oxygen atom of the oxanone ring and the hydroxyl groups of the resorcinol ring, engaging residues Lys334 (2.16 Å), Ser395 (2.50 Å), Ala327 (2.40 Å), Ala391 (1.86 Å), and Gln389 (1.89 Å) of p300. Additionally, a pi–pi interaction is observed with the Hie392 (5.16 Å) of p300 originating from the methoxy resorcinol ring of the compound (Figure 2a).

Similarly, (+) sesamin (EA-46881231) is a lignan compound that has been isolated from the bark and roots of *Zanthoxylum usambarense*, a plant that is used in traditional medicine in Kenya for treating cough, rheumatism, malaria, and other ailments [36]. The (+) sesamin–p300 complex analysis revealed the formation of two hydrogen bonds, and one pi–cation interaction. Both of the hydroxyl groups of the (+) sesamin catechol ring interact via two hydrogen bonds with the Gln398 (1.74 Å, 1.88 Å) of p300. The same catechol ring also interacts with residue Lys334 (4.59 Å) via another pi–cation interaction (Figure 2b). To sum up, both compounds specifically target the cavity of the interface residues, which plays a crucial role in the binding of p300 with HIF-1. These results suggest that these compounds may hinder the interaction between p300 and HIF-1.

### 2.2. Screening of SANPDB to Shortlist Potential Inhibitors of p300-HIF-1 Interaction

In the SANPDB, we identified pedalitin (SA-31161) and kaempferol (SA-5280863) as the phytocompounds with the highest p300 docking scores (−9.580 kcal/mol and −8.179 kcal/mol, respectively) (Table 1). Pedalitin (SA-31161) is a flavonoid compound that has been isolated from the plant *Arctotis arctotoides*, a plant native to Algeria and used in traditional medicine for various purposes [37]. It establishes seven contacts with p300, including six hydrogen bonds and one pi–pi stacking contact. The pedalitin catechol ring interacts with three p300 residues: Thr329, Gln398, and Arg335. The catechol ring hydroxyl groups form hydrogen bonds with Thr329 (1.67 Å), Gln398 (1.90 Å), and Arg335 (2.68 Å, 2.17 Å). The same catechol ring forms pi–pi stacking with Arg335 (4.74 Å). The pedalitin methoxyresorcinol ring interacts with two p300 residues, Gln341 and Hie392. It forms a hydrogen bond with Gln341 (2.31 Å), and pi–pi stacking with the Hie392 (3.58 Å) (Figure 3a).

Kaempferol (SA-5280863) is a flavonoid compound that has been isolated from the plant *Combretum apiculatum*, which is native to Tunisia, Syria, and Egypt and used as an antioxidant [38]. SA-5280863 binds with six interactions with the p300 active site, including four hydrogen bonds and two pi–pi stacking interactions. The hydrogen bonds are formed between the hydroxyl groups of the resorcinol ring and the oxygen atom of the pyranone ring of the compound, with p300 active site residues Ala327 (1.78 Å), Ala391 (1.82 Å), Gln389 (1.92 Å), and Ser395 (2.10 Å). Additionally, pi–cation and pi–pi interactions are formed between the resorcinol ring and residues Lys334 (5.68 Å) and Hie392 (5.18 Å), further stabilizing the interaction with p300 (Figure 3b).

### 2.3. Screening of NEANPD to Shortlist Potential Inhibitors of p300-HIF-1 Interaction

In the case of the NEANPDB, virtual screening against p300 identified the lead compounds as quercetin 3-sulfate (NE-5280362) and (−) epicatechin (NE-72276), with docking scores of −10.287 kcal/mol and −9.017 kcal/mol, respectively (Table 1). Quercetin-3-sulfate (NE-5280362) is a flavonoid compound that has been isolated from the plant *Euphorbia helioscopia*, which is native to Egypt and used in traditional medicine for various ailments, such as skin disease, gonorrhea, migraines, intestinal parasites, and warts [39]. The analysis of the NE-5280362-p300 complex revealed the existence of six interactions, including five hydrogen bonds and one pi–pi stacking interaction. The NE-5280362 sulphate moiety establishes a hydrogen bond with Asp331 (1.33 Å). The catechol and the resorcinol hydroxyl groups extend hydrogen bonds with residues like Ala391 (1.88 Å), Gln389 (1.95 Å), and Ser396 (1.95 Å). p300 Ser395 also establishes a hydrogen bond (2.39 Å) with the pyranone moiety. Similarly, a pi–pi stacking interaction exists between the catechol and resorcinol rings of the ligand and the residue Hie392 (5.48 Å) and Hie392 (5.06 Å) of the p300 active site (Figure 4a).

(−) Epicatechin (NE-72276) is a flavonoid compound that has been isolated from the plant *Helianthemum sessiliflorum*, which is native to Algeria and used in traditional medicine for treating cutaneous lesions. The plant also has an important ecological role in preventing erosion and desertification [40]. NE-72276 forms three hydrogen bonds with the active site of p300. The hydroxyl groups of NE-72276 oxanone and catechol rings establish hydrogen bonds with Lys334 (1.92 Å) and Hie328 (1.69 Å, 2.03 Å) (Figure 4b). Overall, the lead compounds from this database have a strong and specific bonding affinity with the p300 active site.

### 2.4. Screening of NANPDB to Shortlist Potential Inhibitors of p300-HIF-1 Interaction

The virtual drug screening of the NANPDB against p300 identified epicoccolide B (NA-11210533) and farinosone C (NA-11336960) as lead compounds that can block p300-HIF-1 interaction, with docking scores of −10.366 kcal/mol and −7.818 kcal/mol, respectively (Table 1). Epicoccolide B (NA-11210533) is a polyketide compound that has been isolated from the fungus *Epicoccum nigrum*, which is found in various habitats and produces bioactive metabolites [41]. The −10.366 kcal/mol docking score of NA-11210533 shows the stable interaction of this compound with the binding cavity of p300. The analysis of the NA-11210533-p300 complex revealed five hydrogen bonds and one pi–pi interaction. All of the hydrogen bonds were formed through the hydroxyl groups of the trihydroxybenzaldehyde moiety of the ligand and the amino acid residues Gln398 (1.71 Å, 1.69 Å), Gln389 (1.95 Å), and Hie328 (2.24 Å) of p300. A pi–pi interaction was also formed between the dihydroxybenzaldehyde ring and the residue Hie392 (4.38 Å) of p300 (Figure 5a).

Moreover, the farinosone C (NA-11336960) is a polyketide compound that has been isolated from the fungus *Gymnascella dankaliensis*, which is found in desert soils and exhibits cytotoxic activity [42]. Four hydrogen bonds were recorded between the NA-11336960 compound and p300. The three hydroxyls of the methanol and phenol moieties of farinosone C established hydrogen bonds with amino acid residues Hie328 (1.77 Å), Ala394 (2.36 Å), and Ala391 (1.85 Å) of p300. An additional hydrogen bond was also formed between the carbonyl group of the carboxylate region of farinosone C with the p300 residue Ser395 (2.20 Å) (Figure 5b).

### 2.5. Screening of the TCM Database to Shortlist Potential Inhibitors of p300-HIF-1 Interaction

The virtual screening of the TCM database against p300 active site residues identified the two phytocompounds rosmarinic acid (TCM-5281792) and 5-p-trans-coumaroylquinic acid (TCM-6441280), with docking scores of −12.648 kcal/mol and −9.470 kcal/mol, respectively (Table 1). Rosmarinic acid docking scores were the highest among the 10 lead phytocompounds, indicating the highest binding affinity with the binding cavity of p300. Rosmarinic acid (TCM-5281792) is a phenolic compound that has been extracted from the seagrass *Thalassia hemprichii*, which grows in Egypt and exhibits antimicrobial activity [43]. TCM-5281792 interacts with p300 through eight contacts, six of which are hydrogen bonds, a pi–pi interaction, and a salt bridge. The hydrogen bonds were connected to the hydroxyl groups of the catechol and the carboxylate moieties of rosmarinic acid and the Ala391 (1.91 Å), Ser395 (2.26 Å), Gln389 (1.94 Å), Hie392 (2.27 Å), Gln341 (2.08 Å), and Arg335 (1.89 Å) of p300. Additionally, a pi–pi interaction exists between the catechol ring and residue Hie392 (4.74 Å) of p300. The salt bridge between the carboxylate oxygen atom of rosmarinic acid and the Arg335 (4.96 Å) of p300 further stabilizes the interaction by providing electrostatic attraction (Figure 6a).

The second lead compound from TCM is 5-p-trans-coumaroylquinic acid (TCM-6441280), a phenolic compound that has been isolated from the plant *Tribulus terrestris*, which is native to Egypt and has medicinal uses for various conditions, such as urinary disorders, impotence, rheumatism, edema, hypertension, and kidney stones [44]. The TCM-6441280 compound forms a total of seven hydrogen bonds within the p300 binding cavity. Specifically, the hydroxyl and carboxylate groups of the cyclohexane ring of TCM-6441280 interact via hydrogen bonding with Ala327 (2.06 Å), Gln389 (2.33 Å), Ala391 (2.13 Å), Ala394 (2.36 Å), and Ser395 (1.79 Å). Additionally, the carbonyl group in the eastern region of the compound forms a hydrogen bond with the Lys334 (at 2.01 Å) of p300. Lastly, the phenolic hydroxyl group of TCM-6441280 establishes a hydrogen bond with His368 (at 2.17 Å) (Figure 6b).

Our molecular docking analysis suggests that these compounds sourced from various databases exhibit promising potential as inhibitors of the p300 protein. Based on favorable docking scores and target specificity, these compounds are identified as promising candidates for the development of novel inhibitors against p300, which could consequently disrupt its interaction with HIF-1. To further validate our docking results, we conducted molecular dynamics simulations and calculated the binding free energies of the top lead compound from each database.

### 2.6. Dynamics Analysis of the Lead Compound from Each Database and p300

The dynamic stability assessment of the top lead compound from each database and p300 complexes involved calculating the root mean square deviation (RMSD). The analysis of the compound EA-176920, identified from the East African database, in complex with p300, showed that the system initially equilibrated at 2 Å. Subsequently, the RMSD gradually rose to 8 Å at 75 ns before experiencing a sudden drop to 6 Å, maintaining stability until the end of the simulation. After 75 ns the EA-176920-p300 complex showed a higher convergence throughout the simulation period with the average RMSD value of 5.5 Å (Figure 7a). Moreover, the NA-11210533-p300 complex displayed a similar RMSD pattern to the aforementioned complex, albeit with a slightly higher average RMSD. The system stabilized around 4 Å and maintained stability until 125 ns, with minor perturbations observed between 175 and 200 ns. The average RMSD value for this system was approximately 7 Å (Figure 7a). The RMSD analysis of complexes NE-5280362-p300 and SA-31161-p300, shortlisted from the North-East and South African databases, exhibited consistent RMSD values with high convergence throughout the simulation duration. In the case of the NE-5280362-p300 complex, the system stabilized at 40 ns and then maintained the average RMSD of 5 Å throughout the simulation time period (Figure 7a). The SA-31161-p300 complex equilibrated at 2 ns and maintained high convergence until the end of simulation. The average RMSD for this system was 5 Å (Figure 7a). The RMSD of the TCM-5281792-p300 complex exhibited a gradual increased up until 150 ns, after which it stabilized at around 5 Å until 200 ns. The average RMSD for this system was 4 Å (Figure 7a). In total, our analysis revealed that all of the examined compound–p300 complexes exhibited notably reduced RMSD values, indicating robust and stable interactions between the lead compounds and p300. These results underscore the promising therapeutic prospects of these compounds in addressing cancer, specifically through their inhibition of p300.

In addition to the RMSD, we also calculated the root mean square fluctuation (RMSF) for the five lead compound–p300 complexes to check the stability at the residue level. In the field of structural biology, the RMSF is an important tool to study the dynamics of drug–protein interaction [45,46]. The RMSF calculation can reveal how proteins flex and adapt to the presence of drug molecules. By pinpointing flexible regions, the RMSF helps to identify key binding sites for drug design and optimization [47,48]. A decrease in the RMSF value generally corresponds to enhanced stability in the complex, while higher RMSF values indicate increased instability. As shown in Figure 7b, all of the compound–p300 complexes showed a lower RMSF, with most of the residues being in an equilibrium state. Among the lead compounds, the TCM-5281792-p300 complex exhibited the lowest RMSF value, with no fluctuation throughout the simulation period. However, the rest of the complexes showed little fluctuation between amino acids residues 35–45 and 60–70, which might lead to the optimization of interactions and structural conformations. The average RMSF of lead compound–p300 complexes was recorded to be around 2 Å (Figure 7b). This finding indicates that the complexes exhibit a notably high level of stability. Moreover, these results corroborate the data obtained from the RMSD analysis, reinforcing the understanding of the structural integrity and resilience of the complexes.

Furthermore, to check the structural compactness of the five lead phytocompound–p300 complexes, we calculated the radius of gyration (Rg) by using the trajectory of 200 ns simulation. Determining the Rg is paramount for analyzing the dynamic interplay between drugs and proteins. This metric is a cornerstone in deciphering structural alterations and interactions within drug–protein complexes. Quantifying the Rg gives invaluable insights into protein conformational dynamics triggered by drug binding, thereby validating the accuracy of molecular dynamics simulations [49]. Notably, a higher Rg signifies a less compact protein structure, indicative of increased flexibility, whereas a lower Rg denotes enhanced compactness and stability. The analysis of the EA-176920-p300 complex yielded an average Rg value of 14.5 Å until 60 ns. However, a notable fluctuation in the Rg was evident between 60 and140 ns, indicating some structural rearrangements or dynamic behavior within the complex, followed by stability until 200 ns (Figure 8a). The NA-11210533-p300 complex, on the other hand, exhibited a consistent Rg value throughout the simulation period, averaging around 15.5 Å (Figure 8a). Similarly, the NE-5280362-p300 complex demonstrated a comparable Rg pattern, albeit with a slightly lower Rg value, averaging around 14 Å (Figure 8a). Moreover, the SA-31161-p300 complex showed a similar Rg pattern to that observed with the EA-176920-p300 system. The SA-31161-p300 complex Rg remained steady until 90 ns, followed by a noticeable perturbation between 110 and 140 ns (Figure 8a). Likewise, in the TCM-5281792-p300 complex, there was a consistent Rg value throughout the simulation period, albeit with slight deviation observed between 100 and 135 ns. The average Rg for this complex was approximately 14.5 Å (Figure 8a). In conclusion, the resultant lower Rg value of the five lead compound–p300 complexes showcased their structural compactness and further validated the RMSD and RMSF data. Within the domains of computational biology and structural bioinformatics, the evaluation of the solvent accessible surface area (SASA) is significant. This computational method is designed to determine the extent of the surface area of a molecular structure that is accessible to solvent molecules. Widely adopted across scientific disciplines, SASA analysis offers diverse applications in the study of proteins, DNA, RNA, and other biological macromolecules. Its utility extends from deciphering complex molecular interactions to identifying binding sites for ligands, unravelling the nuances of protein–protein interactions, and predicting the behavior of molecules in their interactions with other molecules or their environment [50]. Hence, we performed an SASA analysis of the five lead compounds bound to p300 to assess their level of accessibility. For the EA-176920-p300 complex, the average SASA value was 7500 Å, with slight fluctuations observed at various time points (Figure 8b). Similarly, the NA-11210533-p300 complex maintained a consistent SASA value of around 8000 Å throughout the simulation duration (Figure 8b). The NE-5280362-p300 complex exhibited a pattern similar to EA-176920-p300, with minor fluctuations between 80 and 140 ns, and an average SASA of 7500 Å (Figure 8b). In contrast, the SA-31161-p300 complex displayed a stable SASA value of 7800 Å until 120 ns, followed by a decrease to 7000 Å for the remainder of the simulation, resulting in an average SASA value of 7670 Å (Figure 8b). Moreover, the TCM-5281792-p300 complex maintained a stable SASA value until 100 ns, after which significant fluctuations were observed at different time points throughout the simulation, with an average SASA of 7100 Å (Figure 8b). Changes in SASA values upon drug binding can be correlated with changes in binding affinity. Generally, a decrease in SASA in the region of the binding site may correlate with stronger binding affinity, as it suggests tighter packing of the drug molecule within the binding pocket. In conclusion, the lead compound–p300 complexes showed a consistent SASA value, representing the strong binding affinity of the phytocompounds with the binding cavity of the p300 protein and further validating the results of the molecular docking and RMSD.

To further verify the bonding strength of the five lead compounds with p300, we calculated the average hydrogen bonds formed in each trajectory during the 200 ns dynamic simulation. Calculating the average number of hydrogen bonds in a drug–protein complex post-MD simulation is vital for comprehending binding stability and validating computational models [51]. The analysis offers insights into the strength and dynamics of the drug–protein interactions, guiding the rational design of compounds with improved affinity and selectivity [52]. The hydrogen bonding analysis revealed the average number of hydrogen bonds to be 41, 40, 40, 39, and 43 for EA-176920, NA-11210533, NE-5280362, SA-31161, and TCM-5281792, respectively (Figure 9a–e). The results indicate a strong bonding network between the five lead phytocompounds and p300, suggesting the potential inhibition of p300′s interaction with hypoxia-inducible factor-1 (HIF-1). This inhibition could reduce the aggressiveness of cancer by blocking the hypoxia pathway. These findings align with results from molecular docking studies, as well as RMSD and RMSF analyses, further supporting the potential effectiveness of the five lead compounds in targeting p300-HIF-1 interaction in cancer treatment.

### 2.7. Binding Free Energy and Dissociation Constant (Kd) Analysis of the Five Lead Phytocompound–p300 Complexes

To check the binding strength of lead compounds with the p300 protein, we used the MM/GBSA approach to calculate the binding free energy [53,54]. The calculated van der Waals energies were found to be −31.8393 kcal/mol, −3.7149 kcal/mol, −45.9966 kcal/mol, −41.6263 kcal/mol, and −47.3801 kcal/mol for EA-176920, NA-11210533, NE-5280362, SA-31161, and TCM-5281792, respectively. However, the corresponding electrostatic energies were recorded to be −3.7244 kcal/mol, −1.5064 kcal/mol, −23.7432 kcal/mol, −12.2433 kcal/mol, and −288.4422 kcal/mol, respectively. These results showed the extensive contribution of the van der Waals and electrostatic energies to the bonding strength of the five lead phytocompounds with p300. Moreover, the total binding free energy for EA-176920, NA-11210533, NE-5280362, SA-31161, and TCM-5281792 was found to be −22.0020 kcal/mol, −25.4499 kcal/mol, −32.4530 kcal/mol, −33.9918 kcal/mol, and −57.7755 kcal/mol, respectively (Table 2). Utilizing the PRODIGY approach to calculate Kd values, we further substantiated the robust binding affinity between the lead compounds and p300. Analysis revealed binding scores of −5.34 kcal/mol for EA-176920, −5.39 kcal/mol for NA-11210533, −5.40 kcal/mol for NE-5280362, −5.39 kcal/mol for SA-31161, and −5.42 kcal/mol for TCM-5281792. These scores highlight a strong binding affinity across the protein–ligand complexes, aligning with previous studies that also observed similar ranges of scores [55,56]. However, the binding free energy calculated for each amino acid in the shortlisted compound–p300 complexes is shown in the Appendix A. In summary, the calculated binding free energies showed a notable binding affinity of lead compounds with p300 which could potentially inhibit the binding of p300 with hypoxia-inducible factor-1 and can lead to the inhibition of the hypoxia pathway in cancer. These results further demand for the in vitro and in vivo investigation of the lead compounds.

### 2.8. Oral Bioavailability and Drug Likeness Evaluation of the Five Lead Phytocompounds

To check the drug likeness and oral bioavailability of the five lead compounds, we performed screening according to the Lipinski rule of five. The Lipinski rule of five states that compounds are more likely to be orally active if they have a molecular weight less than 500 Da, a LogP (octanol-water partition coefficient) less than five, no more than five hydrogen bond donors, and no more than 10 hydrogen bond acceptors. Compounds that violate more than one of these criteria may have reduced oral bioavailability [57,58]. The analysis depicted in Table 3 unveiled molecular weights of 320.297, 358.302, 382.302, 316.265, and 360.318 for EA-176920, NA-11210533, NE-5280362, SA-31161, and TCM-5281792, respectively, aligning with the Lipinski criteria. Additionally, all compounds had less than 10 hydrogen acceptors and less than five hydrogen donors, underscoring their potential pharmacological characteristics. Log P values, indicative of a molecule’s lipophilicity and consequential impact on cellular membrane permeability, crucial for absorption and distribution, were recorded as 1.55, 2.86, 1.46, 2.29, and 1.76 for EA-176920, NA-11210533, NE-5280362, SA-31161, and NE-5280362, respectively. Importantly, the analysis of bioavailability indicated a score of 0.55 for EA-176920, NA-11210533, and SA-31161, while scores of 0.11 and 0.56 were observed for NE-5280362 (Table 3). These findings suggest that the five lead compounds hold promise for pharmacological intervention in cancer by inhibiting the interaction between p300 and HIF-1.

### 2.9. ADMET Properties Evaluation of the Five Lead Phytocompounds

Assessing ADMET properties plays a crucial role in mitigating safety risks, enhancing effectiveness, and making informed decisions during drug development [59,60,61]. It is instrumental in obtaining regulatory approval and ensuring the introduction of safe and efficient drugs into the market. By evaluating absorption, distribution, metabolism, excretion, and toxicity early, researchers can pinpoint potential problems, fine-tune drug candidates, and prioritize compounds with advantageous characteristics [62]. Consequently, we assessed the ADMET properties of the five shortlisted phytocompounds. As shown in Table 4, the results of the water solubility analysis indicate that compound NA-11210533 (−3.271) demonstrated the highest water solubility, while that of NE-5280362 (−2.892) was the lowest. Additionally, permeability across the intestinal epithelium was evaluated using the Caco-2 cell model [63], with compound EA-176920 demonstrating the highest permeability compared to others. Efficient uptake by the gastrointestinal tract and systemic distribution are desirable properties for potential drugs. Among the five lead compounds, EA-176920 (60.725%), NA-11210533 (62.577%), and SA-31161 (79.877%) exhibited the highest intestinal absorption, while those of NE-5280362 (30.235%) and TCM-5281792 (32.516%) were the lowest. Furthermore, the volume of distribution (VD) analysis indicated that compounds EA-176920, NA-11210533, and SA-31161 had VD values higher than 0.45, suggesting a preference for tissue distribution; VD values less than −0.15 indicate plasma distribution. Notably, among the five lead phytocompounds, only SA-31161 demonstrated the capability to cross the blood–brain barrier. Toxicity analyses, including AMES toxicity, skin sensitization, and hepatotoxicity, revealed that none of the five compounds exhibited such toxicity effects.

## 3. Materials and Methods

### 3.1. P300 Protein Structure Retrieval and Processing

The 3D structure of molecules is crucial in drug design as it dictates how drugs interact with their target proteins, affecting binding affinity and specificity. Understanding these structural aspects enables the optimization of drug molecules for enhanced therapeutic outcomes while minimizing adverse effects, thus streamlining the drug discovery process [64,65]. In this study, the experimentally determined crystal structure of the p300 protein (ID: 1P4Q) was retrieved from the PBD Protein Data Bank (https://www.rcsb.org/structure/1p4q) (accessed on 8 July 2024) [34]. In the IPQ4 structure, p300 is in complex with the CITED2 transactivation domain; therefore, we used the PyMOL software (2.5.4) to save the p300 protein structure separately. Subsequently, we processed the p300 structure by adding polar hydrogen atoms, removing water molecules, and assigning partial charges. Finally, the structure underwent minimization using the Chimera software (1.15) [66].

### 3.2. Molecular Screening of Drug Databases Against p300

The phytocompounds repositories from different databases such as TCM (Traditional Chinese Medicine), NANPDB (North African Nature Products Database), NEANPDB (North East African Natural Products Database), and SANPDB (South African Natural Products Database) were used for the virtual screening of phytocompounds that bind p300 with high specificity. These phytocompounds, which have varied medicinal properties, were initially processed using the FAF-Drugs4 webserver to select non-toxic, drug-like molecules compliant with Lipinski’s rule of five [67]. The ligands underwent reformatting into pdbqt format, with particular attention paid to the accurate assignment of atomic charges using Open Babel. This included a focus on non-polar hydrogen atoms, Gasteiger charges, and the identification of torsion tree roots for analyzing ligand flexibility. Virtual drug screening was then conducted utilizing EasyDock Vina 2.0, which applies the AUTODOCK4 algorithm to pinpoint and prioritize promising drug-like molecules [68]. Initially, screening was carried out with an exhaustiveness setting of 16 to swiftly identify potential candidates, followed by a more comprehensive screening at an exhaustiveness of 64 to refine results and eliminate false positives. The top 10% of compounds (the range of docking scores was −6.73 kcal/mol to −9.48 kcal/mol) from each database were further analyzed through induced-fit docking (IFD) with AutoDockFR, which allows for receptor flexibility and supports covalent docking, offering rapid and accurate docking results [69]. Finally, the top two compounds from each database were selected based on their favorable docking scores and subjected to dissociation constant determination and molecular dynamics simulation.

### 3.3. Molecular Dynamics Simulation

The molecular dynamics simulation (MDS) aimed to explore the dynamic behavior of proteins in response to inhibitor binding at an atomic scale, using the ff19SB force field and Amber22 package [70]. The complexes were assembled and prepped with the Tleap program, placed within a solvated octahedral box, and each system was neutralized with appropriate counterions (Na^+^ or Cl^−^). System relaxation involved minimizing the energy of each neutralized system in two phases: an initial steep descent method for up to 5000 steps, followed by conjugate gradient optimization. Post-minimization, the systems were gradually heated to 300 K over 50 ps, then equilibrated in two stages at constant pressure (1 atm) and temperature (298 K). This equilibrium phase started with a 50 ps period to stabilize density, followed by a 1 ns period without constraints. The 200 ns production phase was conducted under controlled pressure and temperature conditions, monitored by the Berendsen barostat and Langevin thermostat [71]. Long-range electrostatic forces were computed using the AMBER22 Particle Mesh Ewald (PME) method, with a 10 Å cut-off for van der Waals and electrostatic interactions [72,73,74]. The AMBER22 SHAKE algorithm was applied to refine covalent bonds [75]. All simulations utilized the GPU-accelerated PMEMD.cuda of AMBER22, with trajectory analysis performed using the AMBER22 CPPTRAJ module. Figures were generated by using the Origin Pro Lab v2018.

### 3.4. Dynamics Stability Analysis of Drug–p300 Complexes by Post-Simulation Trajectory

After the simulation, the resultant trajectory data were analyzed using the CPPTRAJ module [76], focusing on root mean square deviation (RMSD), root mean square fluctuation (RMSF), radius of gyration (Rg), and hydrogen bond dynamics [77,78,79]. The RMSD calculation, indicating the average displacement of atoms between the initial and aligned structures, was performed alongside RMSF analysis to assess protein flexibility. To calculate the RMSD, the following formula was used:$$RMSD=\sqrt{\frac{\sum d^{2}i=1}{N_{atoms}}}$$
The RMSF, indicative of residue flexibility, was derived from the B-factor equation, linking it to thermal motion. The following formula was used to calculate the RMSF: Thermal factor or B−factor=[(8π∗∗2)/3] (msf)

The below equation was used to compute the Rg, which evaluates the compactness of the protein structure, reflecting the distribution of mass around the center of gravity.$$R_{gyr}^{2}=\frac{1}{M}\;\sum_{i=1}^{N}m_{i}(r_{i}-R^{2})$$

### 3.5. Kd (Dissociation Constant) Analysis of Shortlisted Compounds and p300 Complexes

Kd analysis holds pivotal importance in drug development owing to its ability to quantify the strength of interactions between drugs and their target proteins [80,81]. This affinity directly impacts the effectiveness of drugs in vivo, as compounds with lower Kd values typically exhibit higher binding affinities and, consequently, enhanced therapeutic potential. Moreover, Kd analysis facilitates the optimization of drug design by enabling the selection of compounds with optimal binding kinetics, thereby improving potency and minimizing off-target effects [82]. Consequently, we calculated the Kd values of the binding of phytocompounds, shortlisted from the virtual screening, to p300 by using the PRODIGY-LIGAND server accessible at https://wenmr.science.uu.nl/prodigy/lig (accessed on 13 September 2024) [83], to evaluate the binding affinity of drug–p300 complexes.

### 3.6. Binding Free Energy Estimation Through MM/GBSA Analysis

The estimation of the binding free energy (BFE) is crucial for understanding how proteins recognize and bind to significant ligands or inhibitors, facilitating the discovery of effective small molecule therapeutics. Utilizing a less resource-intensive and rapid MM/GBSA approach, the BFE for each drug–p300 complex was calculated. The simulation trajectory was used to obtain the stable frame using the MMPBSA.py script to calculate BFE [67]. This involved subtracting the free energy of the bound and unbound states to derive the net binding energy. The comprehensive energy assessment included molecular mechanics contributions, solvation effects, and temperature-related entropy changes, albeit omitting the direct computation of conformational entropy due to its computational demands and potential inaccuracies. The total BFE was calculated by using the following equation:∆Gbind=∆Gcomplex−(∆Greceptor+∆Gligand)
where ΔG(bind) represents the overall binding free energy, and ΔG(complex), ΔG(receptor), and ΔG(ligand) specifically denote the binding free energies associated with the complex, receptor, and ligand, respectively.

However, each component in the total binding free energy was calculated by using the following equation:G=Gbond+Gele+GvdW+Gpol+Gnpol
where Gbond, Gele, GvdW, Gpol, and Gnpol denote the bonded, electrostatic, van der Waal, polar, and non-polar free energy, respectively.

### 3.7. Pharmacokinetics Analysis of Lead Compounds

Adhering to Lipinski’s rule of five, which predicts drug-like characteristics based on molecular parameters, the compounds were evaluated for their oral bioavailability and pharmaceutical potential [84]. Shortlisted compounds were screened for compliance with Lipinski’s criteria using the SwissADME online service (http://www.swissadme.ch/) (accessed on 27 September 2024) [85]. Subsequent in silico ADMET profiling was carried out using the pkCSM platform (https://biosig.lab.uq.edu.au/pkcsm/) (accessed on 27 September 2024) to assess adsorbtion, distribution, metabolism, excretion, and toxicity characteristics, which are pivotal in drug development [86]. This comprehensive analysis aimed to identify compounds with favorable pharmacokinetic profiles, thereby maximizing the chances of success in developing viable drug candidates.

## 4. Conclusions

This study highlights the significant potential of natural phytocompounds as inhibitors of the p300-HIF-1 interaction, providing a novel therapeutic avenue for the treatment of cancer, particularly those cancers reliant on hypoxic adaptations. By screening various databases of natural products, we identified 10 compounds based on their high docking scores with p300. Furthermore, the compound with the highest docking scores from each database (EA-176920, SA-31161, NE-5280362, NA-11210533, and TCM-5281792) was subjected to further analysis, confirming its stable binding affinity with p300 by molecular simulation, binding free energy, and KD analysis. Moreover, the selected compounds followed the Lipinski rule with favorable ADMET properties like efficient intestinal absorption, high water solubility, and no toxicity. The disruption of p300/HIF-1 interaction by these compounds is crucial as it impedes the hypoxia-induced transcriptional activity of HIF-1, thereby potentially stifling tumor progression and angiogenesis. Our research provides strong preliminary evidence supporting the efficacy of these natural inhibitors, paving the way for subsequent in vitro and in vivo testing to validate their therapeutic potential. This approach not only broadens the horizon of cancer therapeutics but also highlights the importance of nature-derived compounds in developing safer, better-targeted, and effective treatment modalities.

## Figures and Tables

**Figure 1 pharmaceuticals-18-00602-f001:**
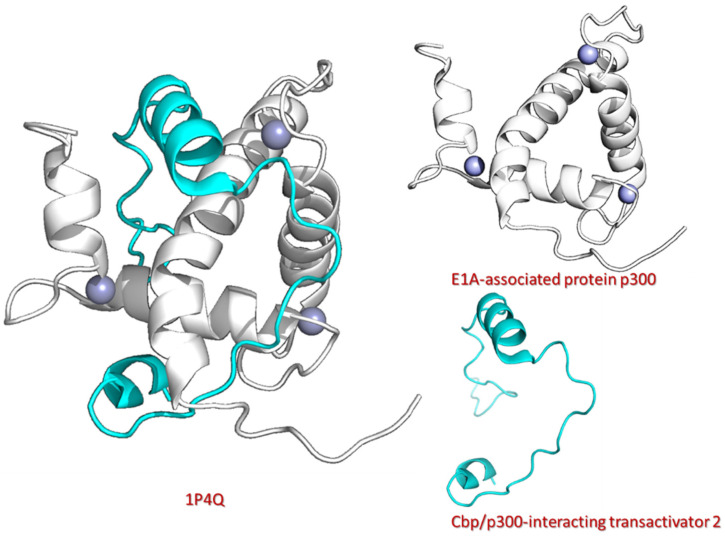
Structure of the CITED2–p300 complex. p300 structure was extracted from the 1P4Q PDB structure using PyMOL.

**Figure 2 pharmaceuticals-18-00602-f002:**
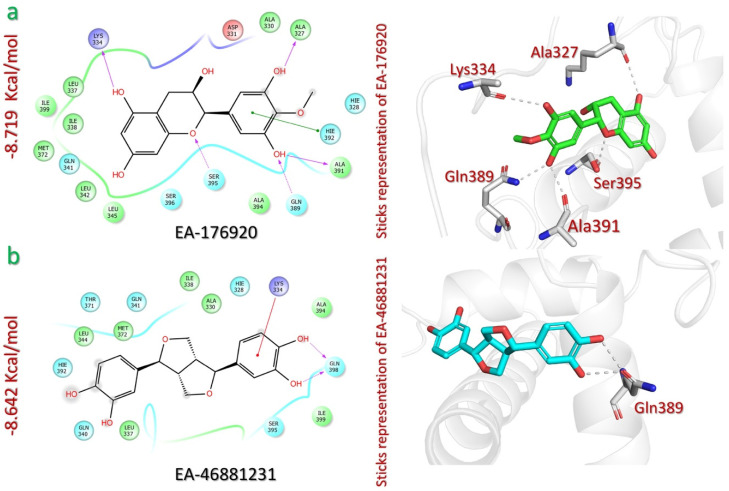
The interaction analysis of EANPDB lead compounds and p300. (**a**) Illustrates the 2D and 3D interaction mechanism of 4′-O-methylepigallocatechin (EA-176920) with the p300 active site. (**b**) Illustrates 2D and 3D interaction mechanism of (+) sesamin (EA-46881231) with the p300 active site. HIE: Histidine with hydrogen on the epsilon nitrogen.

**Figure 3 pharmaceuticals-18-00602-f003:**
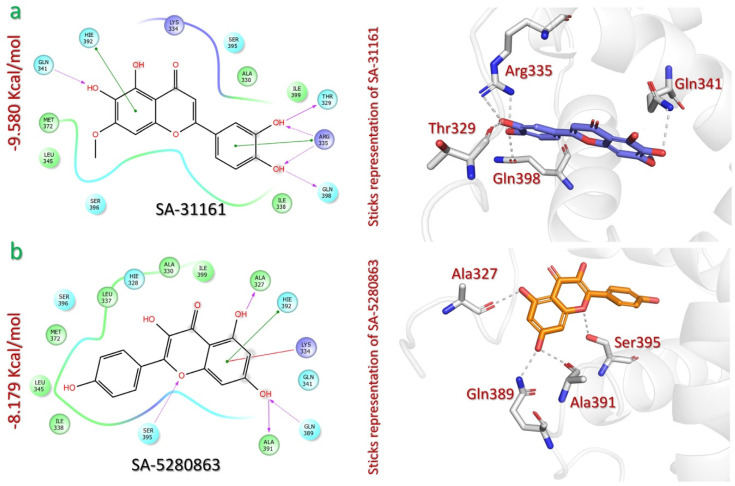
The interaction analysis of SANPDB lead compounds and p300. (**a**) Illustrates the 2D and 3D interaction mechanism of pedalitin (SA-31161) and p300. (**b**) Illustrates the 2D and 3D interaction mechanism of kaempferol (SA-5280863) and p300.

**Figure 4 pharmaceuticals-18-00602-f004:**
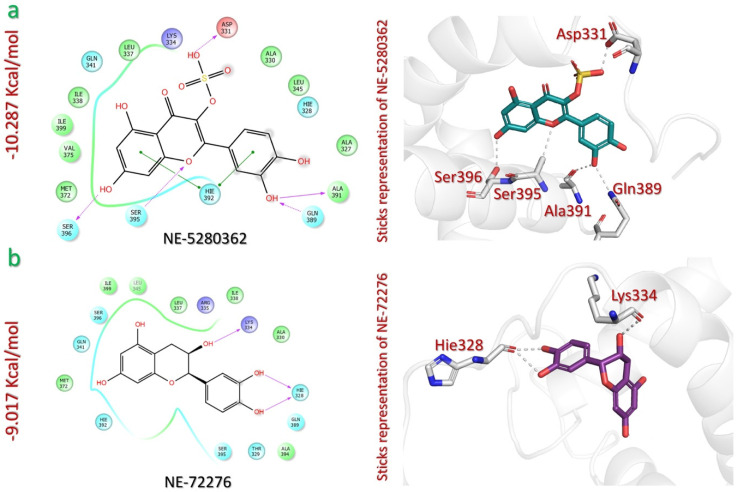
The interaction analysis of NEANPDB lead compounds and p300. (**a**) Illustrates 2D and 3D interaction mechanism of quercetin 3-sulfate (NE-5280362) and p300. (**b**) Illustrates 2D and 3D interaction mechanism of (−) epicatechin (NE-72276) and p300.

**Figure 5 pharmaceuticals-18-00602-f005:**
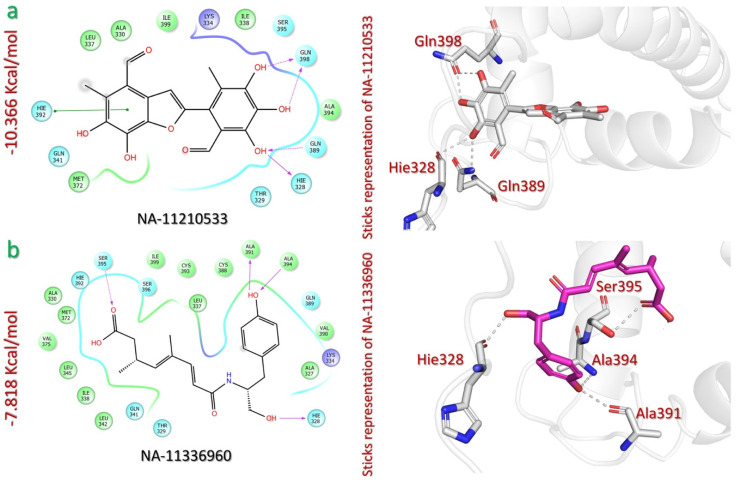
The interaction analysis of NANPDB lead compounds and p300. (**a**) Illustrates 2D and 3D interaction mechanism of epicoccolide B (NA-11210533) and p300. (**b**) Illustrates 2D and 3D interaction mechanism of farinosone C (NA-11336960) and p300.

**Figure 6 pharmaceuticals-18-00602-f006:**
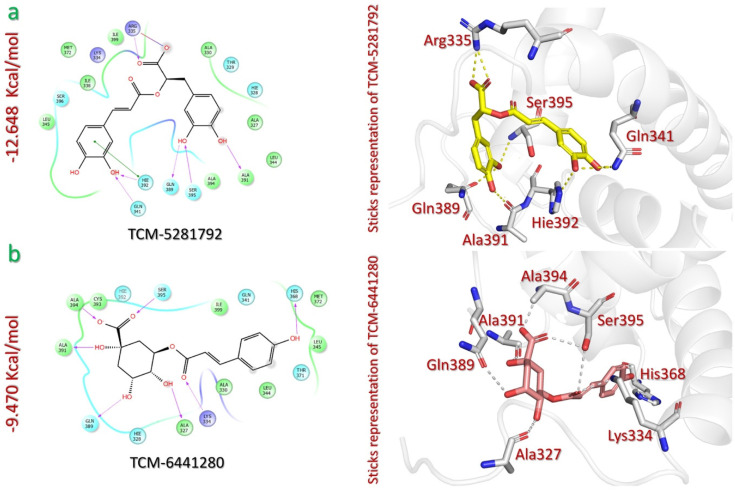
The interaction analysis of TCM lead compounds and p300. (**a**) Illustrates the 2D and 3D interaction mechanism of rosmarinic acid (TCM-5281792) and p300. (**b**) Illustrates the 2D and 3D interaction mechanism of 5-p-trans-coumaroylquinic acid (TCM-6441280) and p300.

**Figure 7 pharmaceuticals-18-00602-f007:**
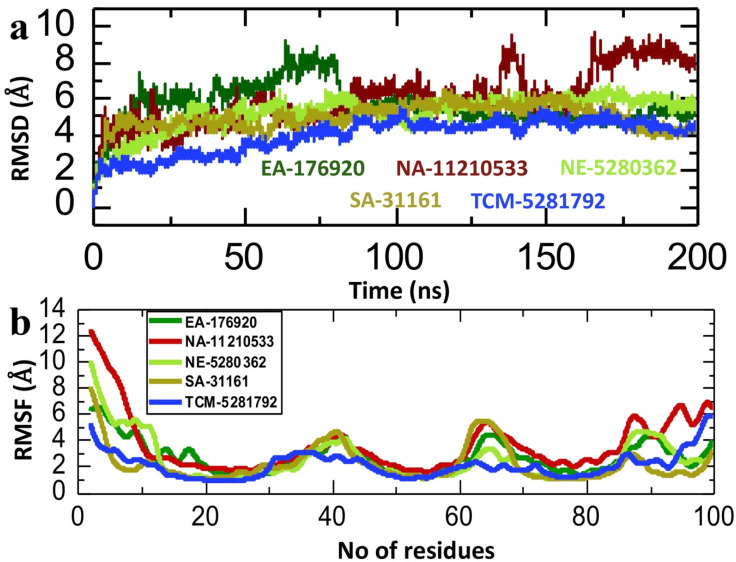
RMSD and RMSF analysis of the shortlisted compound–p300 complexes. (**a**) Illustrates the variations in dynamic stability of shortlisted compounds as RMSD. (**b**) Illustrates the residual fluctuation of shortlisted compounds as RMSF.

**Figure 8 pharmaceuticals-18-00602-f008:**
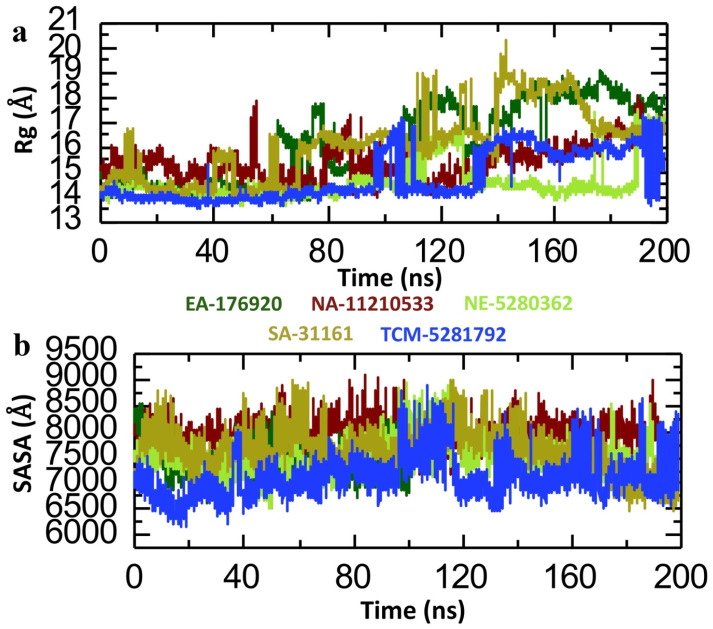
Structural compactness and solvent accessible surface area (SASA) analysis of the five lead compound–p300 complexes. (**a**) Represents the structural compactness of shortlisted compound–p300 complexes as Rg. (**b**) Represents the SASA analysis of shortlisted compound–p300 complexes.

**Figure 9 pharmaceuticals-18-00602-f009:**
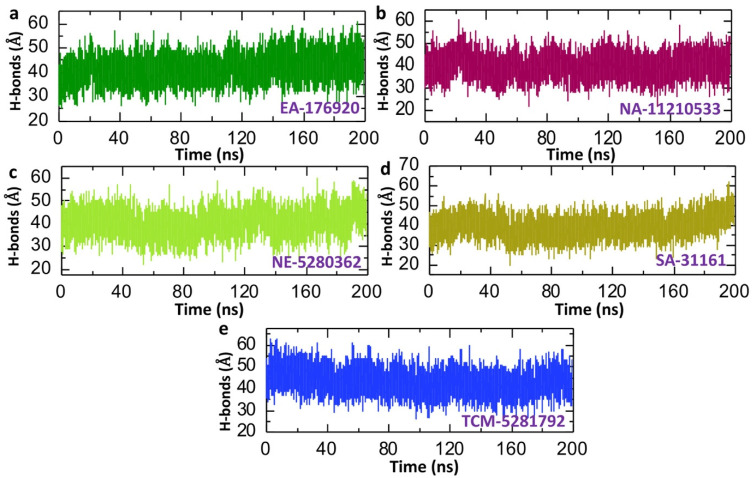
Dynamics of the average number of hydrogen bonds between the five lead phytocompounds and p300: (**a**) shows the average number of hydrogen bonds in the EA-176920-p300 complex; (**b**) shows the average number of hydrogen bonds in the NA-11210533-p300 complex; (**c**) shows the average number of hydrogen bonds in the NE-5280362-p300 complex; (**d**) shows the average number of hydrogen bonds in the SA-31161-p300 complex; and (**e**) shows the average number of hydrogen bonds in the TCM-5281792-p300 complex.

**Table 1 pharmaceuticals-18-00602-t001:** A list of the lead potential inhibitors of p300-HIF-1 interaction phytocompounds from the natural products databases with a detailed analysis of their bonding network with p300.

Database	Compound Name, PubChem ID, and Source	Docking Score	Ligand Atom/Functional Group	P300 (1P4Q)	Interaction	Distance (Å)
**EANPDB**	4′-O-methylepigallocatechin (EA-176920) *Elaeodendron schlechteranum*	−8.719	OH(resorcinol)	Lys334	HB	2.16
			O(oxanol)	Ser395	HB	2.50
			OH(Methoxyresorcinol)	Ala327	HB	2.40
			OH(Methoxyresorcinol)	Ala391	HB	1.86
			OH(Methoxyresorcinol)	Gln389	HB	1.89
			Methoxyresorcinol ring	Hie392	π-π	5.16
	(+)Sesamin(EA-46881231) *Zanthoxylum usambarense*	−8.642	OH(catechol)	Gln398	HB	1.74
			OH(Catechol)	Gln398	HB	1.88
			Catechol ring	Lys334	π-cation	4.59
**SANPDB**	Pedalitin (SA-31161) *Arctotis arctotoides*	−9.580	OH(catechol)	Thr329	HB	1.67
			OH(Catechol)	Gln398	HB	1.90
			OH(catechol)	Arg335	HB	2.68
			OH(Catechol)	Arg335	HB	2.17
			OH(Methoxyresorcinol)	Gln341	HB	2.31
			Methoxyresorcinol ring	Hie392	π-π	3.58
			Catechol ring	Arg335	π-π	4.74
	Kaempferol (SA5280863) *Combretum apiculatum*	−8.179	OH(resorcinol)	Ala327	HB	1.78
			OH(resorcinol)	Ala391	HB	1.82
			OH(resorcinol)	Gln389	HB	1.92
			O(pyranone)	Ser395	HB	2.10
			Resorcinol ring	Lys334	π-Cation	5.68
			Resorcinol ring	Hie392	π-π	5.18
**NEANPDB**	Quercetin 3-sulfate (NE-5280362) *Euphorbia helioscopia*	−10.287	OH(HSO_4_)	Asp331	HB	1.33
			OH(catechol)	Ala391	HB	1.88
			OH(catechol)	Gln389	HB	1.95
			OH(resorcinol)	Ser396	HB	1.95
			O(pyranone)	Ser396	HB	2.39
			Catechol ring	Hie392	π-π	5.48
			Resorcinol ring	Hie392	π-π	5.06
	(−)Epicatechin(NE-72276) *Helianthemum sessiliflorum*	−9.017	O(oxanol)	Lys334	HB	1.92
			OH(catechol)	Hie328	HB	1.69
			OH(catechol)	Hie328	HB	2.03
**NANPDB**	Epicoccolide B (NA-11210533) *Epicoccum nigrum*	−10.366	OH(Trihydroxybenzaldehyde)	Gln398	HB	1.71
			OH(Trihydroxybenzaldehyde)	Gln398	HB	1.69
			OH(Trihydroxybenzaldehyde)	Gln389	HB	1.95
			OH(Trihydroxybenzaldehyde)	Hie328	HB	2.24
			Dihydroxybenzaldehyde ring	Hie392	π-π	4.38
	Farinosone C (NA-11336960) *Gymnascella dankaliensis*	−7.818	OH(methanol)	Hie328	HB	1.77
			OH(phenol)	Ala394	HB	2.36
			OH(phenol)	Ala391	HB	1.85
			CO(COOH)	Ser395	HB	2.20
**TCM**	Rosmarinic acid (TCM-5281792) *Thalassia hemprichii*	−12.648	OH(catechol)	Ala391	HB	1.91
			OH(catechol)	Ser395	HB	2.26
			OH(catechol)	Gln389	HB	1.94
			OH(catechol)	Hie392	HB	2.27
			OH(catechol)	Gln341	HB	2.08
			CO(COOH)	Arg335	HB	1.89
			Catechol ring	Hie392	π-π	4.74
			O(COOH)	Arg335	SB	4.96
	5-p-trans-coumaroylquinic acid (TCM-6441280) *Tribulus terrestris*	−9.470	OH(catechol)	His368	HB	2.17
			Co(easter)	Lys334	HB	2.01
			OH(cyclohexane)	Ala327	HB	2.06
			OH(cyclohexane)	Gln389	HB	2.33
			OH(cyclohexane)	Ala391	HB	2.13
			OH(COOH)	Ala394	HB	2.36
			CO(COOH)	Ser395	HB	1.79

Abbreviations: HB: hydrogen bonds, SB: salt bridge, and Hie: histidine with hydrogen on the epsilon nitrogen.

**Table 2 pharmaceuticals-18-00602-t002:** Binding free energy of the five lead phytocompound–p300 complexes. Abbreviations: ΔEvdw: van der Waals energy change, ΔEele: electrostatic energy change, EGB: generalized Born electrostatic solvation energy, ESURF: non-polar solvation free energy, Delta G Gas: free energy in gas phase, and Delta G Solv: free energy in solvent.

MM/GBSA					
Parameter	EA-176920	NA-11210533	NE-5280362	SA-31161	TCM-5281792
ΔEvdw	−31.8393	−3.7149	−45.9966	−41.6263	−47.3801
ΔEele	−3.7244	−1.5064	−23.7432	−12.2433	−288.4422
EGB	17.7219	−23.5780	42.7170	24.9116	284.4164
ESURF	−4.1602	3.3495	−5.4301	−5.0337	−6.3696
Delta G Gas	−35.5638	−5.2213	−69.7399	−53.8696	−335.8223
Delta G Solv	13.5618	−20.2286	37.2869	19.8778	278.0468
∆G Total	−22.0020	−25.4499	−32.4530	−33.9918	−57.7755

**Table 3 pharmaceuticals-18-00602-t003:** Lipinski rule of five evaluation for the five lead phytocompounds.

Drugs ID	Molecular Weight	Hydrogen Acceptors	Hydrogen Donors	Consensus Log P	Lipinski’s Rule		Bioavailability
					Results	Violation	
EA-176920	320.297	7	5	1.55	Yes	0	0.55
NA-11210533	358.302	8	5	2.86	Yes	0	0.55
NE-5280362	382.302	9	5	1.46	Yes	0	0.11
SA-31161	316.265	7	4	2.29	Yes	0	0.55
TCM-5281792	360.318	7	5	1.76	Yes	0	0.56

**Table 4 pharmaceuticals-18-00602-t004:** ADMET properties of the five lead phytocompounds.

Drugs ID	Water Solubility Log S	Caco-2 Permeability ×10^−6^	Human Intestinal Absorption (%)	VDss (Human)	BBB Permeability	AMES Toxicity	Skin Sensitization	Hepatotoxicity
EA-176920	−3.085	−0.119	60.725	1.635	No	No	No	No
NA-11210533	−3.271	−0.137	62.577	0.482	No	No	No	No
NE-5280362	−2.892	−1.143	30.235	−0.067	No	No	No	No
SA-31161	−3.168	−0.364	79.877	0.661	Yes	No	No	No
TCM-5281792	−3.059	−0.937	32.516	0.393	No	No	No	No

## Data Availability

The original contributions presented in this study are included in the article/Appendix A. Further inquiries can be directed to the corresponding authors.

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
