# Peer review of "Medicinal Phytocompounds as Potential Inhibitors of p300-HIF1α Interaction: A Structure-Based Screening and Molecular Dynamics Simulation Study"

_pharmaceuticals, 2025, doi:10.3390/ph18040602_

Round 1

Reviewer 1 Report

Comments and Suggestions for Authors

Recommendations for Improvement

The reviewer appreciates the authors' work  Medicinal phytocompounds as potential inhibitors of p300-HIF1α interaction: A structure-based screening and molecular 3 dynamics simulation study. However, there are some key points need to be address:

  1. The author provide the interaction of inhibitor against HIF1α based on the computational study is meticulosly presented but author did not provide any concrete conclusion of this interaction. Therefore, it is recommended that author should provide any study reported that plant derived compound exhibited the similar interaction in-vivo or in-vitro. 
  2. Also, it is recommended that author may need to provide Biophysical validation (Binding studies)

By addressing these major revisions, the manuscript will be significantly strengthened, offering greater clarity and broader implications for the scientific community.

Comments on the Quality of English Language

Overall the English Lanquage seems fine but can be improved to retain the readers interest.

Author Response

Response to Reviewers

Dear Editor,

First of all, we are grateful to the editor for the timely response and the reviewers for their useful comments and suggestions to improve our manuscript entitled "Medicinal phytocompounds as potential inhibitors of p300-HIF1α interaction: A structure-based screening and molecular dynamics simulation study" submitted to pharmaceuticals journal, under the manuscript Reference ID: pharmaceuticals-3537506.

We greatly appreciate the referees’ efforts for carefully reviewing our paper and for offering salient suggestions. We found the referees’ feedback very constructive. We tried to be responsive to their concerns and want to extend our appreciation, for taking the time and effort necessary to provide such insightful guidance. 

We believe that we exhaustively resolved all points raised by the referees. We hope that the referees will find our responses to their comments satisfactory, and we are willing to include any future suggestions that the referees might have. 

Below, we respond to each of the comments by the referees, with the comments given in italics followed by our response.

We look forward to working with you and the referees to move this manuscript closer to publication in the Pharmaceuticals journal.

Response to reviewer #1

The reviewer appreciates the authors' work Medicinal phytocompounds as potential inhibitors of p300-HIF1α interaction: A structure-based screening and molecular dynamics simulation study. However, there are some key points need to be address:

  1. The author provide the interaction of inhibitor against HIF1α based on the computational study is meticulosly presented but author did not provide any concrete conclusion of this interaction. Therefore, it is recommended that author should provide any study reported that plant derived compound exhibited the similar interaction in-vivo or in-vitro. 

Response: We thank the reviewer for their valuable suggestion. In response, we have now added supportive literature references to highlight previous studies where plant-derived compounds have shown activity involving the HIF-1α. Specifically, we now cite studies demonstrating that “Several natural and synthetic compounds have demonstrated the ability to modulate HIF-1α signaling, offering promising therapeutic potential against cancer. Chlorogenic acid has been shown to suppress angiogenesis by inhibiting the HIF-1α/AKT pathway, while sinometumine E, derived from Corydalis decumbens, promotes angiogenesis through the HIF-1/VEGF pathway in vivo and in vitro. Graviola, a natural plant-derived compound, has exhibited significant antitumor effects by altering cell metabolism, thereby inhibiting tumorigenicity and metastasis of pancreatic cancer cells. Additionally, chetomin effectively attenuates HIF-1-mediated gene expression, suppressing tumor growth in human colon cancer and PC-3 models. Novobiocin has also emerged as a novel agent that disrupts the interaction between HIF-1α and p300/CBP by directly binding to the HIF-1α C-terminal activation domain.

References added:

Park, J.J., et al., Chlorogenic acid inhibits hypoxia-induced angiogenesis via down-regulation of the HIF-1α/AKT pathway. Cellular oncology, 2015. 38: p. 111-118.

Wang, Y., et al., The natural compound sinometumine E derived from corydalis decumbens promotes angiogenesis by regulating HIF-1/VEGF pathway in vivo and in vitro. Biomedicine & Pharmacotherapy, 2024. 178: p. 117113.

Torres, M.P., et al., Graviola: a novel promising natural-derived drug that inhibits tumorigenicity and metastasis of pancreatic cancer cells in vitro and in vivo through altering cell metabolism. Cancer letters, 2012. 323(1): p. 29-40.

Kung, A.L., et al., Small molecule blockade of transcriptional coactivation of the hypoxia-inducible factor pathway. Cancer cell, 2004. 6(1): p. 33-43.

Wu, D., et al., A novel function of novobiocin: disrupting the interaction of HIF 1α and p300/CBP through direct binding to the HIF1α C-terminal activation domain. PLoS One, 2013. 8(5): p. e62014.

  1. Also, it is recommended that author may need to provide Biophysical validation (Binding studies)

Response: We appreciate the reviewer’s insightful suggestion regarding the inclusion of biophysical validation studies. We acknowledge the importance of experimental bench validation to support in silico findings and confirm their translational potential. We fully understand that such validation is critical for advancing the implications of our research. However, there is two types of studies one is basic and the other is applied. The researchers are working in both fields, in which the basic science is providing the data for the applied science to be validated. Our lab is pure computational which is a separate discipline. We used the latest computational tools such as molecular dynamic simulations, binding free energy calculation and pharmacokinetics analysis whose accuracy is already been tested. Our goal with this submission is to present solid and robust preliminary data generated through structure-based molecular screening and molecular simulation approaches. These findings will serve as a strong foundation for future experimental work and grant applications. We already recommended the experimental validation in the abstract and conclusion section for the future studies.

Reviewer 2 Report

Comments and Suggestions for Authors

Manuscript Title: " Medicinal phytocompounds as potential inhibitors of p300-2 HIF1α interaction: A structure-based screening and molecular 3 dynamics simulation study"

The authors performed virtual screening analyses againt the p300/HIF-1 interaction. The used p300/HIF-1 Traditional Chinese and African Medicine databases for screening. The stability of selected compounds with p300 was confirmed by molecular simulation and  binding free energy analysis. The manuscript is well-written and holds promise for publication in the Pharmaceuticals journal. However, I have a few suggestions to enhance its clarity and impact.

Comments

[1] On page 3, lines 116-117: “In the IPQ4 structure, p300 is in complex with CITED2 transactivation domain and the PyMOL software was used to obtain the structure of p300 protein.” This statement is confusing, as the authors have already mentioned that the p300 structure was downloaded from the PDB database (PDB ID: IPQ4). If the structure was downloaded, how was PyMOL used to obtain the structure? Please clarify.

[2] The authors should explain what cutoff value was used to rank the compounds during the docking simulation.

[3] On page 5, line 223: "Specific p300 residues (Leu-344, Leu-345, Cys-388, 223, and Cys-393) crucial for binding with HIF-1α were targeted." Why were these particular amino acids selected? The authors should discuss their choice and provide relevant references to support their claim.

[4] The authors should provide details on the contribution of significant amino acids to binding with compounds. It would be helpful to include the binding energy for each amino acid.

[5] I recommend that the authors create a single plot for RMSD, RMSF, Rg, etc., for all selected compounds. This would make it easier to compare them.

[6] The figures are of low quality and are difficult to interpret. I request that the authors provide higher-resolution images with clearer labels for better visualization.

Author Response

Response to Reviewers

 Dear Editor,

First of all, we are grateful to the editor for the timely response and the reviewers for their useful comments and suggestions to improve our manuscript entitled "Medicinal phytocompounds as potential inhibitors of p300-HIF1α interaction: A structure-based screening and molecular dynamics simulation study" submitted to pharmaceuticals journal, under the manuscript Reference ID: pharmaceuticals-3537506.

We greatly appreciate the referees’ efforts for carefully reviewing our paper and for offering salient suggestions. We found the referees’ feedback very constructive. We tried to be responsive to their concerns and want to extend our appreciation, for taking the time and effort necessary to provide such insightful guidance. 

We believe that we exhaustively resolved all points raised by the referees. We hope that the referees will find our responses to their comments satisfactory, and we are willing to include any future suggestions that the referees might have. 

Below, we respond to each of the comments by the referees, with the comments given in italics followed by our response.

We look forward to working with you and the referees to move this manuscript closer to publication in the Pharmaceuticals journal.

Response to reviewer #2

The authors performed virtual screening analyses against the p300/HIF-1 interaction. The used p300/HIF-1 Traditional Chinese and African Medicine databases for screening. The stability of selected compounds with p300 was confirmed by molecular simulation and binding free energy analysis. The manuscript is well-written and holds promise for publication in the Pharmaceuticals journal. However, I have a few suggestions to enhance its clarity and impact.

Comments

[1] On page 3, lines 116-117: “In the IPQ4 structure, p300 is in complex with CITED2 transactivation domain and the PyMOL software was used to obtain the structure of p300 protein.” This statement is confusing, as the authors have already mentioned that the p300 structure was downloaded from the PDB database (PDB ID: IPQ4). If the structure was downloaded, how was PyMOL used to obtain the structure? Please clarify.

Response: Thank you for your comment. We retrieved the p300-CITED2 complex from the PDB database (ID: IPQ4). Using PyMOL, we separated the structures of p300 and CITED2 to isolate the p300 structure for further processes such as drug screening. We have now rephrased the sentence to avoid any confusion.

[2] The authors should explain what cutoff value was used to rank the compounds during the docking simulation.

Response: Thank you for the comment. As we mentioned in the methodology, from the initial screening of databases we selected the top 10% compounds (the range of docking score was -6.73 kcal/mol to -9.48 kcal/mol) and subjected these compounds for the induced-fit docking protocol. After the induced-fit docking we selected two compounds from each database on the basis of high docking score. 

[3] On page 5, line 223: "Specific p300 residues (Leu-344, Leu-345, Cys-388, 223, and Cys-393) crucial for binding with HIF-1α were targeted." Why were these particular amino acids selected? The authors should discuss their choice and provide relevant references to support their claim.

Response: Thank you for your comment. The aim of our study is to inhibit the interaction between p300 and HIF1α; therefore, we specifically targeted the p300 residues involved in this interaction. A previous study titled “Structural basis for recruitment of CBP/p300 by hypoxia-inducible factor-1” identified Leu-344, Leu-345, Cys-388, and Cys-393 of p300 as crucial residues for binding with HIF1α. Following your suggestion, we have added the reference and rephrased the sentence to avoid any confusion. The following reference has been added in the manuscript.

Freedman, S.J., et al., Structural basis for recruitment of CBP/p300 by hypoxia-inducible factor-1α. Proceedings of the National Academy of Sciences, 2002. 99(8): p. 5367-5372.

[4] The authors should provide details on the contribution of significant amino acids to binding with compounds. It would be helpful to include the binding energy for each amino acid.

Response: Thank you for your very valuable comment. According to your suggestion we calculated the binding energy for each amino acid and included in the manuscript as supplementary figure 1.

[5] I recommend that the authors create a single plot for RMSD, RMSF, Rg, etc., for all selected compounds. This would make it easier to compare them.

Response: We appreciate the reviewer’s suggestion to consolidate the RMSD, RMSF, Rg, and other dynamic parameters into single comparative plots. Following your recommendation, we have combined the figures for RMSD, RMSF, Rg, and SASA. However, we opted to keep the hydrogen bonds figure separate, as the hydrogen bond ranges for the selected compound-p300 complexes are quite similar. Combining them would make it challenging to distinguish and interpret the data for each compound. 

[6] The figures are of low quality and are difficult to interpret. I request that the authors provide higher-resolution images with clearer labels for better visualization.

Response: We thank the reviewer for their observation. According to your suggestion all figures were prepared and exported at a resolution of 600 dpi, in accordance with the journal's submission guidelines to ensure high-quality visualization. Labels, fonts, and figure elements were carefully formatted to maintain clarity at publication standards.

Reviewer 3 Report

Comments and Suggestions for Authors

This manuscript by Suleman et al. deals with the significant potential of natural phytocompounds as inhibitors of the p300-HIF-1 interaction, while a novel therapeutic avenue for the treatment of cancer, particularly those reliant on hypoxic adaptations was presented. The results shows the identification of 10 compounds based on screening various databases of natural products. Topic is intersting and deserves publication in the journal. I would recommend its publication after following issues are addressed.

  1. Lines 18-36 ".... : The pathological role of hypoxia in cancer progression, notably through the stabilization and activation of hypoxia-inducible factor-1 (HIF-1), is well-documented.........In this study, we identified potential com-pounds with high docking scores such as EA-176920 (-8.719), EA-46881231(-8.642), SA-31161 (- 9.580), SA-5280863 (-8.179), NE-5280362 (-10.287), NE-72276 (-9.017), NA-11210533 (-10.366), NA- 11336960 (-7.818), TCM-5281792 (-12.648) and TCM-6441280 (-9.470 kcal/mol) as lead compounds."  Abstract section is not clear regarding what is going on the study.
  2. Lines 94-101. "...This study seeks to uncover novel inhibitors targeting the p300 enzyme by exploring a wide range of natural product databases. These databases, including the Chinese Medicines Database (TCM), North African Natural Products Database (NANPDB), East African Natural Products Database (EANPDB), North-East African Natural Products Database (ANPDB), and the South African Natural Compounds Database (SANCDB), comprise natural products sourced from various origins, including plants, animals, fungi, bacteria, and marine organisms."   this part needs refs.
  3. Line 116, the address regarding protein databank says nothing. 
  4. Line 211, it should be "adsorbtion"
  5. Figure 1 and Table 1 need more explanation in the text. 

Author Response

Response to Reviewers

 Dear Editor,

First of all, we are grateful to the editor for the timely response and the reviewers for their useful comments and suggestions to improve our manuscript entitled "Medicinal phytocompounds as potential inhibitors of p300-HIF1α interaction: A structure-based screening and molecular dynamics simulation study" submitted to pharmaceuticals journal, under the manuscript Reference ID: pharmaceuticals-3537506.

We greatly appreciate the referees’ efforts for carefully reviewing our paper and for offering salient suggestions. We found the referees’ feedback very constructive. We tried to be responsive to their concerns and want to extend our appreciation, for taking the time and effort necessary to provide such insightful guidance. 

We believe that we exhaustively resolved all points raised by the referees. We hope that the referees will find our responses to their comments satisfactory, and we are willing to include any future suggestions that the referees might have. 

Below, we respond to each of the comments by the referees, with the comments given in italics followed by our response.

We look forward to working with you and the referees to move this manuscript closer to publication in the Pharmaceuticals journal.

Response to reviewer #3

This manuscript by Suleman et al. deals with the significant potential of natural phytocompounds as inhibitors of the p300-HIF-1 interaction, while a novel therapeutic avenue for the treatment of cancer, particularly those reliant on hypoxic adaptations was presented. The results show the identification of 10 compounds based on screening various databases of natural products. Topic is intersting and deserves publication in the journal. I would recommend its publication after following issues are addressed.

  1. Lines 18-36 ".... : The pathological role of hypoxia in cancer progression, notably through the stabilization and activation of hypoxia-inducible factor-1 (HIF-1), is well-documented.........In this study, we identified potential com-pounds with high docking scores such as EA-176920 (-8.719), EA-46881231(-8.642), SA-31161 (- 9.580), SA-5280863 (-8.179), NE-5280362 (-10.287), NE-72276 (-9.017), NA-11210533 (-10.366), NA- 11336960 (-7.818), TCM-5281792 (-12.648) and TCM-6441280 (-9.470 kcal/mol) as lead compounds."  Abstract section is not clear regarding what is going on the study.

Response: We appreciate your helpful comment regarding the clarity of the abstract. In response, we have thoroughly revised the abstract to provide a clearer and more concise overview of the study objectives, computational methods, key results, and the potential implications of our findings. The updated abstract now better reflects the structure and scope of the study and highlights the significance of the identified phytocompounds in targeting the p300–HIF-1α interaction.

  1. Lines 94-101. "...This study seeks to uncover novel inhibitors targeting the p300 enzyme by exploring a wide range of natural product databases. These databases, including the Chinese Medicines Database (TCM), North African Natural Products Database (NANPDB), East African Natural Products Database (EANPDB), North-East African Natural Products Database (ANPDB), and the South African Natural Compounds Database (SANCDB), comprise natural products sourced from various origins, including plants, animals, fungi, bacteria, and marine organisms."   this part needs refs.

Response: Thank you for the comments. According to your suggestion the following references have been incorporated in the mentioned section of manuscript.

Wang, Y., et al., A critical assessment of traditional Chinese medicine databases as a source for drug discovery. Frontiers in Pharmacology, 2024. 15: p. 1303693.

Fidele, N.-K., et al., NANPDB: A Resource for Natural Products from Northern African Sources. 2017.

Simoben, C.V., et al., Pharmacoinformatic investigation of medicinal plants from East Africa. Molecular informatics, 2020. 39(11): p. 2000163.

Suleman, M., et al., Molecular screening of phytocompounds targeting the interface between influenza A NS1 and TRIM25 to enhance host immune responses. Journal of Infection and Public Health, 2024. 17(7): p. 102448.

Diallo, B.N.t., et al., SANCDB: an update on South African natural compounds and their readily available analogs. Journal of Cheminformatics, 2021. 13(1): p. 37.

  1. Line 116, the address regarding protein databank says nothing. 

Response: We thank you for pointing this out. We would like to clarify that the Protein Data Bank (PDB) is a globally recognized and widely used repository for the three-dimensional structural data of biological macromolecules. It serves as the standard reference for retrieving experimentally determined protein structures for use in computational and structural biology research. Accordingly, the p300 structure (PDB ID: 1P4Q) used in our study was obtained from this publicly accessible and well-established database. You can check by clicking or pasting the below web link in the browser.

https://www.rcsb.org/structure/1p4q

  1. Line 211, it should be "adsorbtion"

Response: Thank you for the comment. The correction has been incorporated.  

  1. Figure 1 and Table 1 need more explanation in the text. 

Response: Thank you for the comment. According to your suggestion, the Figure 1 and Table explained in detail and the following paragraphs have been included in the manuscript.

“The figure 1 illustrates the structural organization of p300, highlighting its key functional regions. The left panel shows the complex of p300-CITED2, with secondary structural elements depicted. The right panels further depict the E1A-associated protein p300 and the CBP/p300-interacting transactivator 2, emphasizing the domains involved in protein-protein interactions.”

“Table 1 summarizes the docking analysis of the top 10 lead compounds identified from natural product databases as potential inhibitors of the p300-HIF-1 interaction. The table presents docking scores, key ligand functional groups, and their interactions with p300, including hydrogen bonding, hydrophobic interactions, and π-π stacking. The interaction distances (in Å) highlight binding strength and stability. Notably, the compounds NE-5280362, NA-11210533 and TCM-5281792 exhibited the strongest binding, forming multiple hydrogen bonds with key residues. These findings suggest promising candidates for further validation through molecular dynamics simulations and experimental studies.”

Round 2

Reviewer 1 Report

Comments and Suggestions for Authors

Can be Accepted with minor corrections

Comments on the Quality of English Language

Need to be improve

Reviewer 3 Report

Comments and Suggestions for Authors

The authos addressed all my cocerns and it is now 

Author Response

Comment: The authors addressed all my concerns, and it is now for Authors 

Response: Thank you for considering our study for publication.